# Synthesis of Low Sidelobe Pattern with Enhanced Axial Radiation for Sparse Conformal Arrays Based on MCDE Algorithm

**Ning Zhang [1], Zhenghui Xue [2,*], Pei Zheng [3], Lu Gao [3] and Jiaqi Liu [3]**

[1] EMC and Microwave System Laboratory, Beijing Institute of Technology, Beijing 100081, China
[2] School of Integrated Circuits and Electronics, Beijing Institute of Technology, Beijing 100081, China
[3] National Key Laboratory of Science and Technology on Test Physics and Numerical Mathematics, Beijing 100076, China
* Correspondence: zhxue@bit.edu.cn

**Abstract:** A hybrid optimization method for the synthesis of a sparse conformal array with the verification of a truncated cone antenna array is proposed in this manuscript. This array synthesis is studied aiming at enhancing axial radiation and reducing the peak sidelobe level (PSLL) by figuring out the optimal antenna element arrangement and corresponding feeding scheme. A multi-agent composite differential evolution (MCDE) algorithm is established by integrating a multi-agent system (MAS) with a differential evolution (DE) algorithm. In addition, a hybrid strategy method and a time-varying weighting factor are added to the mutation operator to accelerate convergence. Two examples of 64-element and 900-element truncated cone antenna arrays were synthesized. After forming a sparse antenna array out of the original full array, the number of antennas was decreased to 80% and 56.8%, respectively. The results show that the main beam of the sparse conformal antenna array is accurately fixed to the axial direction with the PSLL less than $-20$ dB at both the $\varphi = 0°$ and $\varphi = 90°$ planes, which proves the effectiveness of this method in conformal sparse array synthesis.

**Keywords:** sparse array 1; conformal antenna 2; differential evolution (DE) 3; multi-agent system (MAS) 4

## 1. Introduction

Due to the capacity of conforming to surfaces, conformal antennas have been successfully applied in many scenarios in recent decades [1]. A truncated cone, as an aerodynamic structure, has been applied in many streamlined aircrafts such as missiles and rockets. However, for this truncated cone structure, especially in the case of a small cone angle, two issues will come into view. First, it is difficult to install a large amount of electronic equipment such as transceivers and feeding networks in the narrow space of the cone tip, which makes it impossible to deploy a large-scale antenna array. In addition, the small cone angle proposes an arduous challenge to the performance of axial radiation, which has a great impact on the missile's data transmission. Despite the above difficulties, several studies have been proposed in recent years due to the wide application of truncated cones [2–6].

The sparse array can maintain or even improve the radiation characteristics with fewer antenna elements and can reduce installation space for electronic equipment by arranging the antenna elements nonuniformly, which is very applicable in a truncated cone. However, unlike sparse linear and planar arrays, which already have many mature synthesis methods [7–13], such as iterative numerical methods and evolutionary algorithms (EAs), sparse conformal array synthesis has not been solved effectively due to its complex structure. Some effort has been made to solve specific problems in sparse conformal array synthesis [14–24], such as analytical techniques [14], compressive sensing (CS) techniques [15],

distributed aperture synthesis [16,17], and evolutionary algorithms [5,18–22]. In [23], a versatile multi-task Bayesian compressive sensing (MT-BCS) strategy was used to match a given reference pattern based on a nonuniform conformal array. However, the existing research on sparse conformal arrays mainly focuses on array synthesis, rather than reducing the number of antenna elements. Therefore, array synthesis to achieve a sparse array with low PSLL is of great research significance.

The aforementioned works mainly studied suppression of the sidelobes. For the truncated cone array, fixed axial radiation is also of great importance. Some novel antenna elements were designed to enhance the axial radiation in [25,26]. A four-element slot array with parasitical dual U-shaped slots in the axial direction is proposed in [25]; however, this antenna element is not suitable for large antenna arrays due to its undesirable sidelobe level. At the same time, some optimizing algorithms for conformal surfaces have been proposed to make the main beam point in a specified direction [27–29]. Concerning both low sidelobe levels and the axial radiation, a brain storm optimization (BSO) method was developed in [30]. However, the proposed array achieves a good radiation pattern with axial radiation on a spherical surface instead of a truncated cone array. It is an urgent necessity to study the sparse array synthesis based on the truncated cone conformal array that realizes both a low sidelobe pattern and enhanced axial radiation.

Inspired by natural biological evolution, the evolutionary algorithm optimizes the objects by using different operators, such as the crossover operator and mutation operator. Compared with traditional calculus-based methods, the evolutionary algorithm is a mature global optimization method with high robustness and wide applicability. Moreover, owing to the characteristics of self-organization, self-adaptation, and self-learning, it can effectively deal with complex problems that are difficult for traditional optimization algorithms. Evolutionary algorithms have been widely used in antenna array optimization in recent decades, including genetic algorithms (GA) [30], particle swarm optimization (PSO) algorithms [31], grey wolf optimization (GWO) [32], invasive weed optimization (IWO) [33], ant colony optimization (ACO) [34], biogeography-based optimization (BBO) [35], the backtracking search optimization algorithm (BSA) [30], etc.

In recent years, the application of GA to antenna array synthesis has been widely studied. In [36], Yan and Lu suppressed the peak SLL of linear antenna arrays by optimizing the excitation current, whereas Tian and Qian [37] added the positions of elements as a variable and improved the performance by optimizing both variables simultaneously. Furthermore, the fitness function was redesigned to enhance the global searching property of GA. In 2006, Chen et al. [38] proposed a modified real genetic algorithm (MGA), which improved the optimization efficiency of the algorithm by reducing the search space. As an improvement to the GA, the differential evolution (DE) algorithm is also widely used in the field of array synthesis. S. Yang et al. [39] applied the DE algorithm to the synthesis of linear arrays, and an element rotation technique was proposed later in [40]. Moreover, the DE/best/1/bin strategy was adopted to realize the shaping of power patterns, which broaden the application scope of the DE algorithm [41]. Based on the above research, this paper continues the study of the optimization of conformal sparse arrays.

The DE algorithm has the advantage of strong applicability [42]. The addition of a multi-agent system (MAS) can overcome many EA's drawbacks such as slow iteration speeds and the production of a wide range of applications in many fields [43,44]. However, to the best of our knowledge, research on the integration of MAS and DE algorithms has rarely been seen in the field of electromagnetics. Since we expect to obtain an optimal antenna element arrangement and corresponding feeding scheme, so as to generate axial radiation with the lowest PSLL, which is a multi-dimensional nonlinear problem with multi-constraints, the concept of MAS is then applied to the DE algorithm to this end. In addition, mutation–selection strategies and time-varying weighting factors are adopted to accelerate the convergence of the algorithm. The obtained multi-agent composite differential evolution (MCDE) algorithm has proved its superiority with an optimization example of 64 array elements ending with an 80% sparse rate. To test the performance of the proposed algorithm

in synthesizing the array on a large scale, an example with 900 antenna elements was simulated and compared with the benchmark algorithm.

The MCDE algorithm is described in Section 2. The details of the chosen antenna element and the simulation example are shown in Section 3. The fabrication analysis is analyzed in Section 4. Finally, the conclusion is summarized in Section 5.

## 2. MCDE Algorithm

### 2.1. Array Formulation

As shown in Figure 1, in the rectangular coordinate system, a truncated cone array is formed with the origin as the center of the bottom surface. Assuming that $N_a$ antenna elements are uniformly arranged at the initial moment, and N antenna elements are evenly placed on each layer with M layers in total. There are $N_{full}$ antenna elements when the element spacing is $\lambda/2$, where $\lambda$ is the electromagnetic wavelength in the free space. As $\gamma_0$ is the preset sparse rate, $N_a = N_{full} \cdot \gamma_0$. The height of the truncated cone is H, and the half cone angle is $\theta_C$. The antenna element can be denoted as $(m, n)$, and $I_{mn}$ is the corresponding current excitation. All the antenna elements are hypothesized to be identical, i.e., $I_{mn} = 1$ for all elements. The array factor AF can be expressed as equation (1).

$$AF = \sum_{m=0}^{M} \sum_{n=0}^{N} I_{mn} \times e^{jkr_{mn}\psi_{mn}} \tag{1}$$

where $k = 2\pi/\lambda$, $r_{mn}$ is the distance between the antenna element $(m, n)$ and the origin of the coordinate. The phase of the antenna element $\psi_{mn}$ can be expressed as: $\psi_{mn} = sin\theta_m(sin\theta cos(\varphi - \varphi_{mn}) - sin\theta_0 cos(\varphi_0 - \varphi_{mn})) + cos\theta_m(cos\theta - cos\theta_0)$. $\theta_0$ and $\varphi_0$ are the main beam directions. In order to suppress the PSLL, the sum of PSLL at $\varphi = 0°$ and $\varphi = 90°$ planes is selected as the fitness function. $\theta_{mn}$ and $h_{mn}$ are the angle and height of the element $(m, n)$, respectively, the objective function and constraint conditions are as follows:

$$f(x) = min\left\{ \left| \frac{AF(\theta, 0°)}{FF_{max}} \right| + \left| \frac{AF(\theta, 90°)}{FF_{max}} \right| \right\}$$

$$s.t.\ 0 \leq m \leq M, 0 \leq n \leq N \tag{2}$$
$$0 \leq \theta_{mn} \leq 2\pi, 0 \leq h_{mn} \leq H, \theta_{min} \leq \theta$$
$$mn - \theta_{m(n-1)}, \frac{\lambda}{2} \leq h_{mn} - h_{(m-1)n}$$

where $FF_{max}$ represents the peak gain of the main lobe, and $\theta_{min} = 45\lambda/(k_{mn}\pi^2)$ is the minimum spacing between two adjacent elements. Next, the optimization algorithm can be applied to this array model to realize the pattern synthesis of the truncated cone conformal array.

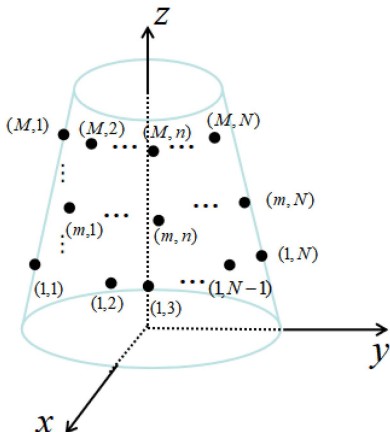

**Figure 1.** The geometry of a truncated cone sparse array.

### 2.2. Algorithm Descriptions

In MCDE, P rows and Q columns of agents form a lattice-like environment with each agent containing $N_a$ parts. The $(p, q)$ agent can be expressed as $\left\{ \vec{x}_{p,q} = (x_{p,q,1}, x_{p,q,2}, \cdots x_{p,q,N_a}), p = 1, 2, \ldots, P, \ q = 1, 2, \ldots, Q \right\}$. In this environment, the agent $\vec{x}_{p,q}$ can only exchange information with its neighboring agents, and its neighborhood is called:

$$N_{p,q} = \left\{ \vec{x}_{p+1,q}, \vec{x}_{p-1,q}, \vec{x}_{p,q+1}, \vec{x}_{p,q-1} \right\} \tag{3}$$

As for the agent located on the boundary, its neighborhood is defined as the boundary agent on the other side.

The algorithm flow is as follows:

(1)   Initialization:

Generate random individuals in an N-dimensional environment space, and the formula for each individual is as follows:

$$\vec{x}_{p,q} = \vec{x}_{p,q}^{\,min} + rand(0,1) \times (\vec{x}_{p,q}^{\,max} - \vec{x}_{p,q}^{\,min})$$
$$p = 1, 2, \ldots, P; q = 1, 2, \ldots Q \tag{4}$$

where, *rand* (0,1) is a uniformly distributed random variable between [0, 1], $\vec{x}_{p,q}^{\,min}$ and $\vec{x}_{p,q}^{\,max}$ are the maximum and minimum search boundary, $L_{size}$ is the population.

(2)   Neighborhood competition:

Calculates the fitness value of each agent in the neighborhood, and compares the fitness value with the agent that has the highest one.

$$f(\vec{x}_{p,q}) > f(M_{p,q}) \tag{5}$$

Define $M_{m,n}$ as the agent with the highest fitness value in the neighborhood of $\vec{x}_{p,q}$. If the above formula is met, $\vec{x}_{p,q}$ can be retained, otherwise, the new agent is selected according to the following rules:

$$\vec{x}'_{p,q} = M_{p,q} + rand(0,1) \times (\vec{x}_{p,q} - M_{p,q}) \tag{6}$$

(3)   DE operator

DE algorithm is used to optimize the entire environment's agents. In order to facilitate the optimization of DE, the order of the agent is changed from the previous $(p, q)$ to $i$, thus $\left\{ \vec{x}_{i,G} = (x_{i,1,G}, x_{i,2,G}, \cdots x_{i,N_a,G}), G = 1, 2, \ldots, G_{\max}, \right\}$, where $N_a$ is the population. The DE operator establishes a mutation vector $v$ for the target vector $\vec{v}_{r1,G} = (v_{i,1,G}, v_{i,2,G}, \cdots v_{i,2,G})$, as the penalty function can effectively transform the non-linear optimization problem into an unconstrained one. Therefore, the sum of the agent's penalty function $P(x)$ can be calculated as:

$$P(x) = f(x) + \left\lceil e^{\alpha(1-\rho)\frac{G}{G_{max}}} - 1 \right\rceil g(x) \tag{7}$$

where $P(x)$ is set as decreasing function, $\alpha$ is an adjustable function, usually an integer between [0, 5], $\rho$ is the ratio of feasible solutions between neighbors, $g(x)$ is the designated penalty item. The proportion of feasible solutions increases through an iterative process, whereas the penalty gradually decreases. This helps the algorithm to gradually shift from searching for feasible solutions to qualified solutions. In particular, since the MCDE's population is small, a time-varying weight factor $G/G_{\max}$ is added to make sure the function changes relatively smoothly.

CoDE, the composite DE algorithm [45], combines several effective trial vector generation strategies to improve the performance of DE. Here, in the MCDE algorithm, three different mutation strategies combined with a binomial crossover strategy are performed to obtain three trial vectors $\vec{u}_{i,G}^{1}$, $\vec{u}_{i,G}^{2}$ and $\vec{u}_{i,G}^{3}$ with randomly selected control parameters, and the results with the best fitness are chosen as the final vector. "DE/rand/1/bin" has strong global search capabilities; conversely, "DE/best/1/bin" reserves strong local search capabilities by mutating on the optimal individual. To maintain the population diversity in the early stage and improve the convergence ability in the late stage, the third mutation strategy is designed as a combination of the first two strategies and weighted by a time-varying weight factor.

Mutation strategies combined with binomial crossover:

DE/rand/1/bin

$$u_{i,j,G}^{1} = \begin{cases} x_{r1,j,G} + F \cdot \left( x_{r2,j,G} - x_{r3,j,G} \right) \ if \ rand \leq \ CR \ or \ G = k \\ x_{i,j,G}, otherwise \end{cases} \tag{8}$$

DE/best/1/bin

$$u_{i,j,G}^{2} = \begin{cases} x_{\text{best},j,G} + F \cdot \left( x_{r2,j,G} - x_{r3,j,G} \right) \ if \ rand \leq \ CR \ or \ G = k \\ x_{i,j,G}, otherwise \end{cases} \tag{9}$$

DE/rand-best/1/bin

$$u_{i,j,G}^{3} = \omega \cdot u_{i,j,G}^{2} + (1 - \omega) \cdot u_{i,j,G}^{1} \tag{10}$$

where $i = 1, 2, \ldots, P \cdot Q$, $j = 1, 2, \ldots, N_a$, $k$ is *a* number randomly selected in $[1, N_a]$, $x_{best,j,G}$ is the jth element of the best individual, $x_{r1,j,G}$, $x_{r2,j,G}$, $x_{r3,j,G}$ are the jth elements of other individuals randomly selected in the population. CR and F are the crossover rate and the scaling factor, respectively. Time-varying weighting factor $\omega$ changes according to the following formula:

$$\omega = \omega_{\text{min}} + (\omega_{\text{max}} - \omega_{\text{min}}) \cdot \frac{G}{G_{\text{max}}} \tag{11}$$

where $\omega_{max}$ and $\omega_{min}$ are the maximum and minimum weight values, respectively, and $(\omega_{\text{max}}, \omega_{\text{min}}) \in (0,1)$, $G$ and $G_{max}$ are the current and the preset maximum iteration number. It can be seen that the time-varying weighting factor gradually increases through an iterative process, and the third mutation strategy also accelerates to obtain the optimal solution.

The selection operator is used to select the better ones from the target vector $\vec{x}_{i,G}$ and trial vector $\vec{u}_{i,G}$ for the next generation:

$$\vec{x}_{i,G+1} = \begin{cases} \vec{u}_{i,G}, if \ f\left( \vec{u}_{i,G} \right) \leq f\left( \vec{x}_{i,G} \right) \\ \vec{x}_{i,G}, otherwise \end{cases} \tag{12}$$

(4)　Self-learning operator

Finally, a self-learning operator extracted from MAS is used to accelerate convergence, which is performed only on the optimal agent to alleviate the cost of calculation. Generate a new agent environment, where the agent $\left\{ \vec{x}_{p,q} = (x_{p,q,1}, x_{p,q,2}, \cdots x_{p,q,N_a}), p = 1, 2, \ldots, P, q = 1, 2, \ldots, Q \right\}$ is produced by:

$$\vec{x}_{i,G+1} = \begin{cases} \vec{x}_{p,q,G}, p = 1, q = 1 \\ \vec{x}_{p,q,G}^{new}, otherwise \end{cases} \tag{13}$$

where $\overrightarrow{x}_{p,q}^{\text{new}} = (x_{p,q,1}, x_{p,q,2}, \cdots x_{p,q,N_a})$ is generated by the following formula:

$$x_{p,q,j} = \begin{cases} x_{p,q}^{\min}, \overrightarrow{x}_{p,q} \cdot r_1 < x_{p,q}^{\min} \\ x_{p,q}^{\max}, \overrightarrow{x}_{p,q} \cdot r_1 > x_{p,q}^{\max} \\ \overrightarrow{x}_{p,q,G} \cdot r_1, otherwise \end{cases} \tag{14}$$

where $j = 1, 2, \ldots, N_a$, $r_1 = rand(1 - R, 1 + R)$ and $R \in [0, 1]$ represents the search radius. The self-learning of the agent is completed through the above method.

The flowchart for array synthesis with nonequal placement using the MCDE algorithm is shown in Figure 2. The active radiation pattern obtained from the full-wave simulation is used as the element antenna's active radiation pattern, which contributes to the reliability of the optimization. In order to improve the search efficiency of the algorithm, the difference in the distance between two adjacent elements is selected, so that the search area in the algorithm can be reduced. Therefore, an initial population whose individuals are the difference in the distances of adjacent units is first generated, and the elements of each individual are sorted from smallest to largest. Then, the individual values are transformed into the true antenna element spacing. Next, the optimization is performed in a loop using the aforementioned MCDE algorithm, and the termination condition is judged at each generation until the condition is satisfied.

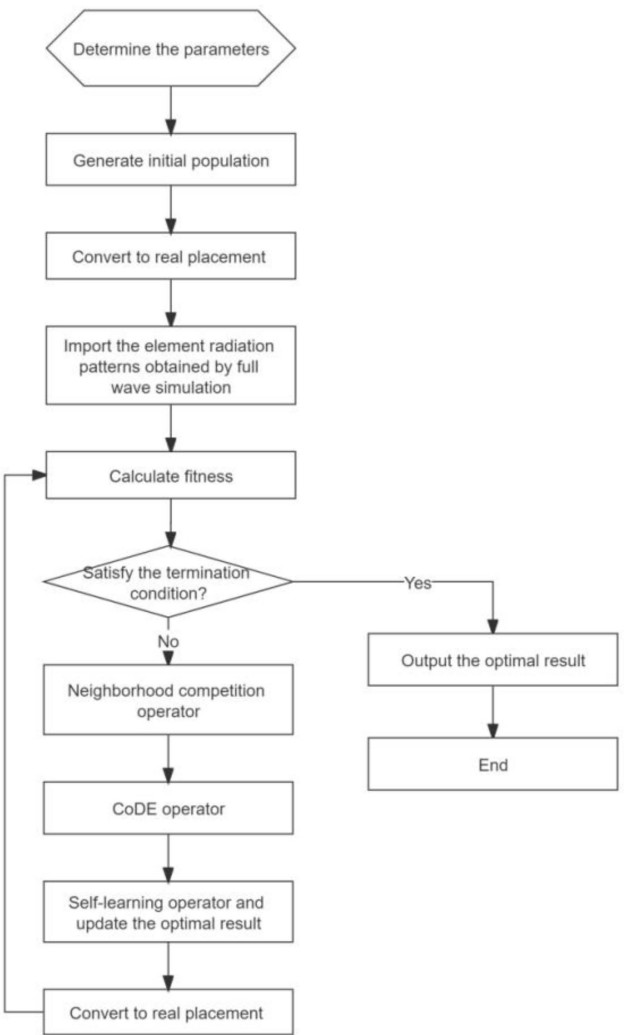

**Figure 2.** The flow chart of antenna array placement optimization using the MCDE algorithm.

## 3. Antenna Element Design and Simulation Results

As shown in Figure 3, the proposed dual-polarized patch antenna is composed of three substrate layers and one air layer, and Rogers 5880 ($\varepsilon_r$ = 2.2) is used as a substrate in this antenna. The patch is etched with a U-shaped groove to achieve miniaturization while broadening the beamwidth. The microstrips with a length of L2 placed on two sides of the patch are connected to the SMP connector to excite the patch, thus radiating dual-polarization electromagnetic waves. Detailed dimensions of the proposed antenna geometry are summarized in Table 1.

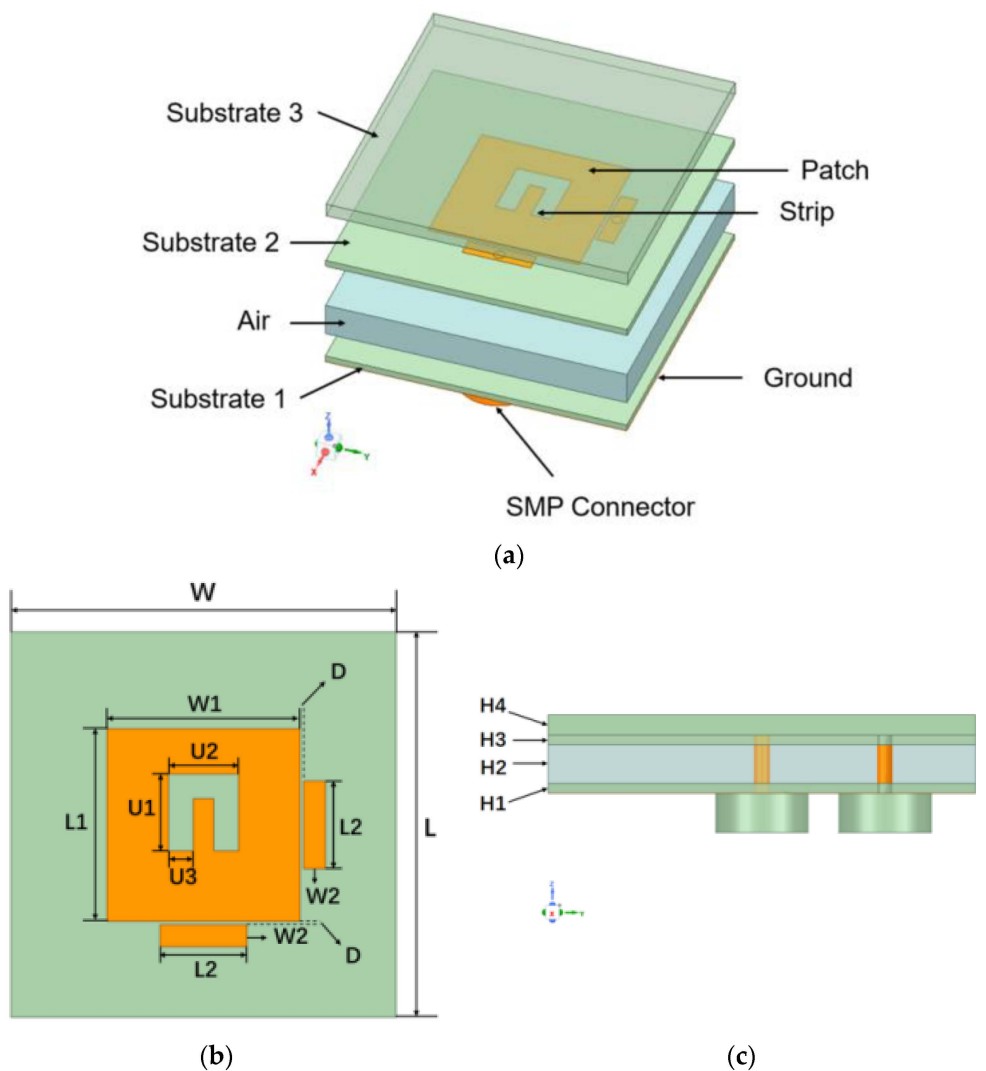

**Figure 3.** Views of the proposed antenna. (**a**) Perspective view. (**b**) Top view. (**c**) Side view.

**Table 1.** Dimensions for the proposed antenna.

| $W, L$ | $L_1, W_1$ | $L_2$ | $W_2$ | $D$ | $H_1, H_1$ |
|---|---|---|---|---|---|
| 9 | 5.5 | 2.5 | 0.63 | 0.1 | 0.254 |
| $H_2$ | $H_4$ | $U_1$ | $U_2$ | $U_3$ | |
| 1 | 0.508 | 1.5 | 2 | 0.7 | |

The antenna operates with a voltage standing wave ratio (VSWR) less than 1.7 in the operating bandwidth of 15–17 GHz (Figure 4a). As shown in Figure 4b, good similarity is observed in the radiation pattern of two planes with a boresight gain of 6.5 dBi. The half-power beam widths (HPBWs) in the $\varphi = 0°$ plane and $\varphi = 90°$ plane are 84° and 86°,

respectively, which contributes to axial scanning. Due to the half cone angle of a truncated cone array, the axial radiation can be regarded as a wide-angle scan of the conformal array, which can be achieved by an antenna element with wide HPBW.

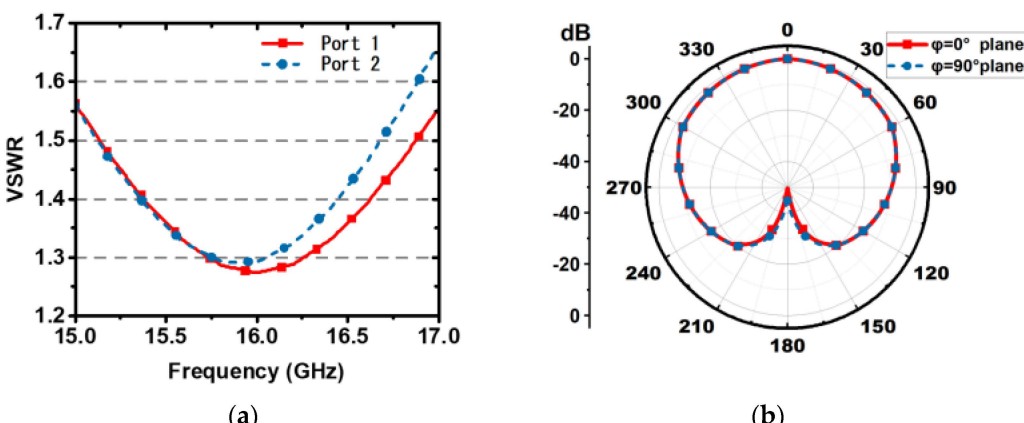

(a)                    (b)

**Figure 4.** Simulated results of the antenna element in HFSS. (**a**) Simulated VSWRs at Port 1 and Port 2. (**b**) Simulated radiation patterns in the $\varphi = 0°$ and $\varphi = 90°$ planes at 16 GHz.

The implemented sparse conformal antenna is shown in Figure 5 for better understanding; 27 aforementioned elements are arranged on the side surface of the cone based on the optimized positions. Since each element has a specific disposing-phase to achieve the axial radiation, phase shifters are needed for each element.

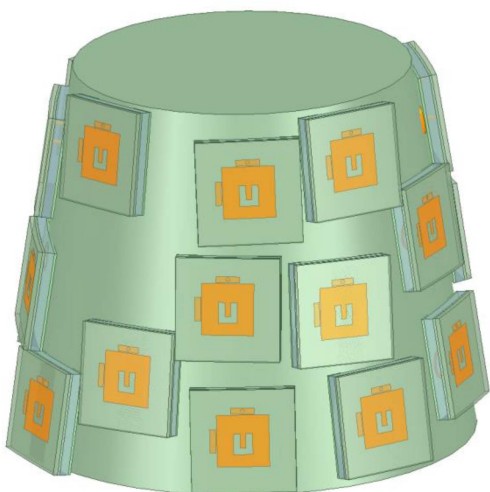

**Figure 5.** An example of implemented antenna array.

The pattern synthesis of arrays in the previous literature generally treats the element as the ideal omnidirectional antenna and ignores the coupling between elements, which will reduce the optimization time. However, simplification in these two aspects will lead to a large difference between the actual array and the simulated one. In order to solve the problems above, the active element pattern is theoretically verified in [46]. In the case of an infinite antenna array where each element is excited, the active element pattern contains the coupling effects. Therefore, using the active element pattern can effectively improve the accuracy of the array synthesis. In this research, the antenna element's active radiation pattern obtained from the full-wave simulation in HFSS is imported into MATLAB to include coupling effects.

MATLAB 2016 on a PC operating at 3.6 GHz with 16 GB of RAM is used for the simulation of MCDE and other benchmark algorithms. For MCDE optimization, the P and

Q are taken as 7. Therefore, the size of the population is 49. The generation is set as 200, scaling factor F = 0.85, and crossover rate CR = 0.9. To reduce the computational time for self-learning operator, size sL =3, sGen = 5, and sR = 0.25 are taken.

The benchmark algorithms used in this paper are DE [44], GA [31], PSO [32], GWO [33], and IWO [34]. All algorithms have the same population size, which is 49. The parameters CR and F are set as 0.9 and 0.85 in DE. GA has a crossover probability of 0.7 and a mutation probability of 0.1. For PSO, the cognitive component, social component, and inertia weight are set to be 2, 1, and 2, respectively. GWO includes a control parameter that linearly decreases from 2 to 0. The minimum and the maximum number of seed is 0 and 5, respectively, with a modulation index of 2 in IWO. The initial standard deviation and the final standard deviation are set as 0.01 and 0.1, respectively.

### 3.1. Example 1: Pattern Synthesis of 52-Element Array

An array with $N_a$ = 52 is established to verify the proposed synthesis method. The parameters of the surface are shown in Table 2. The DE, GA, PSO, GWO, IWO, and the new MCDE algorithms are applied to synthesize the array and the radiation patterns are compared with the full array result. The geometry of the full array and sparse array obtained by the MCDE algorithm are shown in Figure 6 for comparison. It can be seen that the elements are unevenly distributed after optimization, thus achieving better performance than uniform arrays in some aspects.

**Table 2.** The parameters of the example array.

| $N$ | $M$ | $N_{full}$ | $H(\lambda)$ | $r_n$ | $r_0$ | $\theta_C$ | $\gamma_0$ | $(\theta_0, \varphi_0)$ |
|---|---|---|---|---|---|---|---|---|
| 13 | 4 | 64 | 5.5 | 2 | 0.63 | 30 | 80% | $(0, 0)$ |

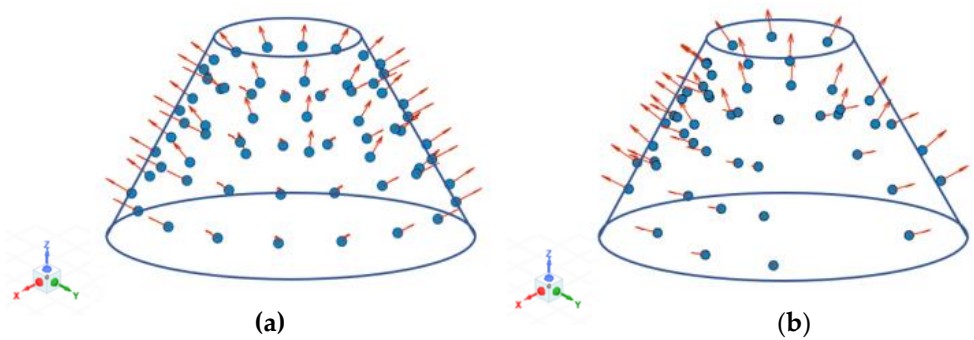

|                (a)                |                (b)                |

**Figure 6.** The geometry of the full array and sparse array. (**a**) Full array. (**b**) Sparse array.

The final arrangement of the 52-element array after MCDE optimization is shown in Table 3, where one column represents one layer of the array. In each blank, the front number represents the height of each antenna element($\lambda$) in the cone and the latter means the angle of the element in the cylindrical coordinate system (°).

The convergence plot of all optimization algorithms is shown in Figure 7. It can be seen that the MCDE algorithm attains a PSLL of about $-20.84$ dB within 50 iterations, whereas the final PSLL obtained by the benchmark algorithm is above $-16$ dB after 100 iterations, showing the capacities of fast convergence, stability, and the small required population.

The proposed array is installed on a truncated cone surface instead of a flat plane, so the main beam is not directed to the array axis ($0°$, $0°$) when the elements are all fed in equal-amplitude and in-phase. Therefore, axial radiation is achieved thorough phase disposing. The normalized radiation patterns ($0° < \theta < 180°$) of the full array and sparse array synthesized by MCDE and other benchmark algorithms are shown in Figure 7. It can be seen that the main lobe synthesized by any algorithm can accurately scan to ($0°$, $0°$) with high symmetry along $\theta = 0°$.

**Table 3.** The Optimal Placement of the Example Array (Unit: mm).

| Lane 1 | Lane 2 | Lane 3 | Lane 4 |
|---|---|---|---|
| 1.17, 0.03 | 0.97, 1.61 | 1.25, 32.62 | 0.71, 5.85 |
| 0.28, 34.03 | 0.31, 46.93 | 0.00, 61.67 | 1.97, 46.36 |
| 0.10, 66.17 | 1.33, 71.59 | 0.75, 84.05 | 0.16, 75.31 |
| 1.36, 81.77 | 0.03, 116.42 | 1.80, 112.37 | 0.50, 115.40 |
| 1.21, 98.56 | 0.23, 145.50 | 1.36, 144.37 | 1.62, 144.96 |
| 0.43, 113.96 | 1.90, 168.79 | 0.75, 179.70 | 0.16, 176.58 |
| 0.81, 147.32 | 1.95, 131.71i | 1.26, −149.98 | 1.06, 130.88 |
| 1.25, −168.58 | 0.06, −99.33 | 0.48, −75.47 | 1.60, −23.28 |
| 1.11, −104.21 | 0.98, −63.87 | 1.14, −25.41 | 1.47, 9.81 |
| 0.25, −78.28 | 1.72, −39.53 | 1.96, −0.77 | 0.28, 53.58 |
| 0.33, −62.88 | 0.48, −13.93 | 1.69, +39.78 | 0.87, 103.69 |
| 0.00, −18.19 | 1.66, 4.30 | 0.56, +68.90 | 0.70, 132.65 |
| 0.83, 2.93 | 1.62, 23.19 | 1.36, +92.87 | 0.95, 173.48 |

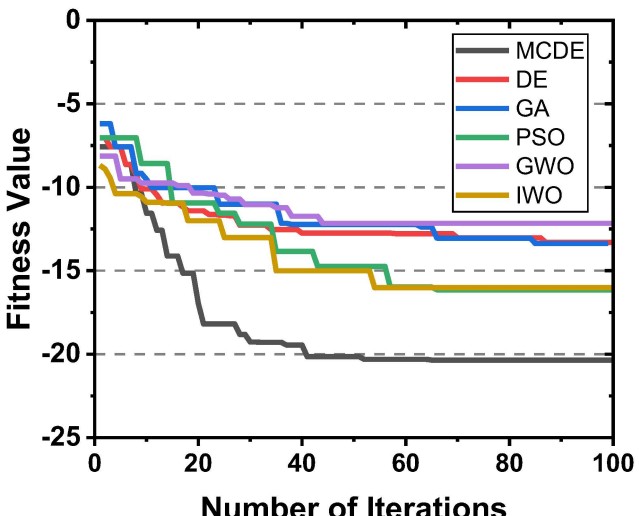

**Figure 7.** Iterations for the MCDE and benchmark algorithms.

The performances of the sparse array after different optimizations and the full array are detailed in Table 4, including PSLL, directivity, gain, and beamwidth. Compared with the PSLL = −9.97 dB of the full array, a certain reduction is achieved by different optimizations. The comparison are shown in Figure 8. Among them, MCDE achieves −20.84 dB and −21.53 dB in the $\varphi = 0°$ and $90°$ planes, respectively, which are better than the PSO result of −16.30 dB. For directivity, it is worth emphasizing that the gain of the array is theoretically reduced to 21.55 dBi due to the number of elements being reduced to 80%. Therefore, only the IWO and the proposed MCDE algorithm achieve gains larger than the theoretical one of 0.02 dBi and 0.32 dBi, which verify the enhancement of the axial gain. The gain of the array is calculated by:

$$G = 10\lg(D \cdot E_a) \tag{15}$$

where $D$ is the directivity obtained by array optimization, and $E_a$ is the efficiency of the proposed dual-polarized patch antenna at the corresponding frequency, which is 85% at 16 GHz. Since the beam width has a negative correlation with the aperture of the array, the HPBW of the array will remain unchanged if the aperture is maintained after optimization. Different from the IWO algorithm that narrows the beam, the MCDE algorithm can keep the HPBW similar to that of the full array.

**Table 4.** The Performances of the Array I.

|  | PSLL (dB) | Directivity (dBi) | Gain (dBi) | HPBW (°) |
|---|---|---|---|---|
| Full Array | −9.97 | 22.52 | 21.81 | 15 |
| DE | −13.70 | 21.23 | 20.52 | 14 |
| GA | −13.87 | 21.15 | 20.44 | 12 |
| PSO | −16.30 | 20.89 | 20.18 | 12 |
| GWO | −12.16 | 21.48 | 20.77 | 12 |
| IWO | −16.04 | 21.57 | 20.63 | 11 |
| MCDE (Proposed) | −20.84 | 21.87 | 20.92 | 15 |

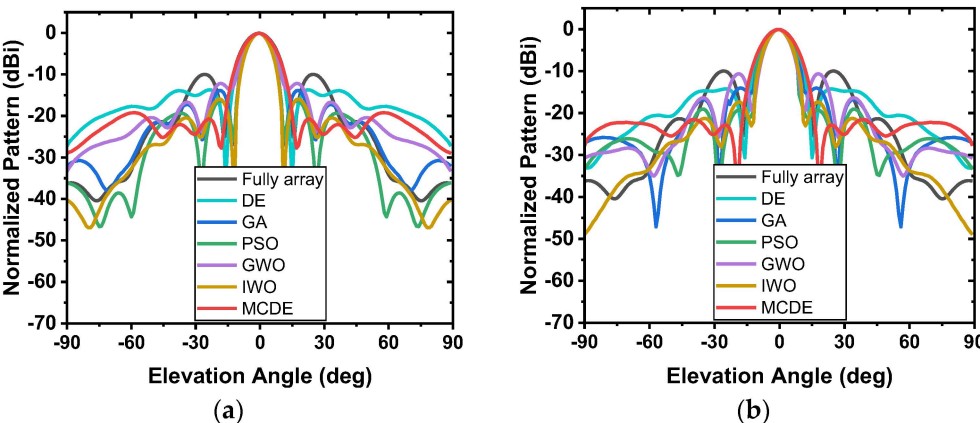

**Figure 8.** Radiation patterns of the full array, sparse array synthesized by MCDE and DE algorithm. (**a**) φ = 0° plane. (**b**) φ = 90° plane.

In conclusion, the MCDE algorithm shows certain competitiveness in terms of PSLL and gain compared with other optimization algorithms. The 3D radiation pattern of the sparse array synthesized by the MCDE algorithm is shown in Figure 9 to reveal more details of the array's radiation.

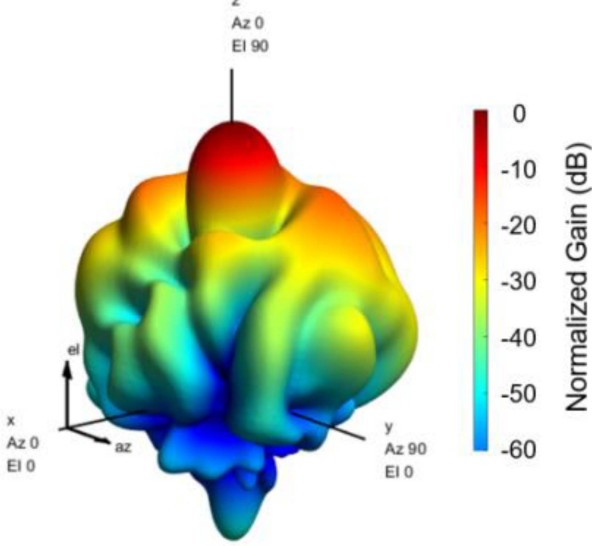

**Figure 9.** The 3D radiation pattern of the sparse array synthesized by MCDE algorithm.

*3.2. Example 2: Pattern Synthesis of 512-Element Array*

The large-scale array optimization problem will challenge the stability and the efficiency of the optimization algorithm. To further investigate the optimization capability

of the MCDE algorithm for large-scale arrays, a 512-element sparse array based on the truncated cone surface was built, in which the sparsity was set to 56.8%, i.e., 900 antenna elements existed in the array before sparsity. The specific metrics of the array are shown in Table 5. Similar to Example 1, a dual-polarized patch antenna is used as the array element, and its active radiation pattern obtained from full-wave simulation is imported for synthesis. Again, the objective of the array optimization is to achieve a fixed main beam to $(0°, 0°)$ and the lowest PSLL. Likewise, the present large-scale array is regarded as a phased array for beam steering. The sparse array layout obtained from the MCDE optimization is shown in Figure 10.

**Table 5.** The Parameters of the Example Array 2.

| $N$ | $M$ | $N_{full}$ | $H(\lambda)$ | $r_n$ | $r_0$ | $\theta_C$ | $\gamma_0$ | $(\theta_0, \varphi_0)$ |
|-----|-----|-----------|-------------|-------|-------|-----------|-----------|------------------------|
| 32  | 16  | 900       | 7.5         | 3.5   | 1.5   | 15        | 56.8%     | $(0,0)$                |

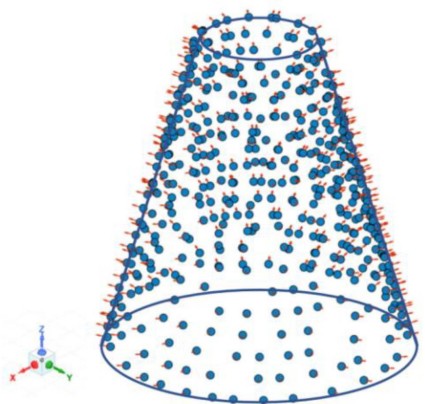

**Figure 10.** The geometry of sparse array example 2.

The radiation pattern of the full array and the sparse array obtained by the benchmark algorithms are shown in Figure 11. It can be seen that the arrays after different optimizations can still achieve axial and symmetrical radiation in large-scale cases.

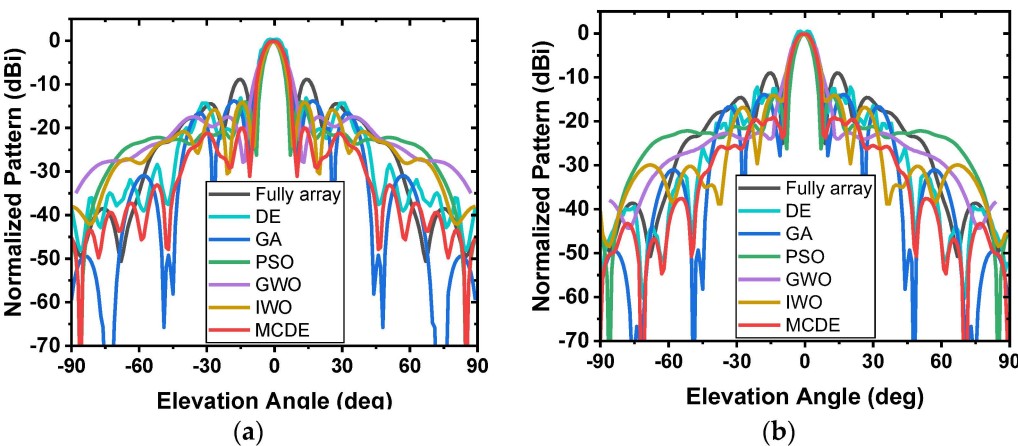

**Figure 11.** Radiation patterns of the full array, sparse array synthesized by MCDE and DE algorithm. (**a**) $\varphi = 0°$ plane. (**b**) $\varphi = 90°$ plane.

The array performances obtained by full arrays and sparse arrays synthesized by different algorithms are shown in Table 6, including PSLL, directivity, gain, and HPBW. Compared with the PSLL of the full array, all the sparse arrays achieved reductions of more than 4 dB, especially for the GWO and MCDE, which are 6.38 dB and 12.23 dB, respectively.

As a reference for comparison, the theoretical gain of the array after elements reduction is calculated to be 29.98 dBi, while the full array achieves a directivity of 32.44 dBi. The gains of arrays synthesized by the IWO and MCDE algorithms are 0.04 and 0.16 dB higher than the theoretical one, respectively. Other algorithms can hardly guarantee the HPBW, while the array optimized by the MCDE algorithm maintains the HPBW the same as the full array. It can be seen that in the case of nearly 1000 elements before synthesis with a sparsity of 56.8%, the algorithm still shows stable performance, achieving the goals of both a large number of elements and large sparsity. The three-dimensional radiation pattern of the array is shown in Figure 12.

**Table 6.** The Performances of the Array 2.

|  | PSLL (dB) | Directivity (dBi) | Gain (dBi) | HPBW (°) |
|---|---|---|---|---|
| Full Array | −8.85 | 32.44 | 31.73 | 9 |
| DE | −13.12 | 28.82 | 28.11 | 11 |
| GA | −13.77 | 29.12 | 28.41 | 9 |
| PSO | −14.04 | 29.75 | 29.04 | 7 |
| GWO | −15.23 | 29.45 | 28.74 | 8 |
| IWO | −14.11 | 30.02 | 29.31 | 8 |
| MCDE (Proposed) | −21.08 | 30.14 | 29.43 | 9 |

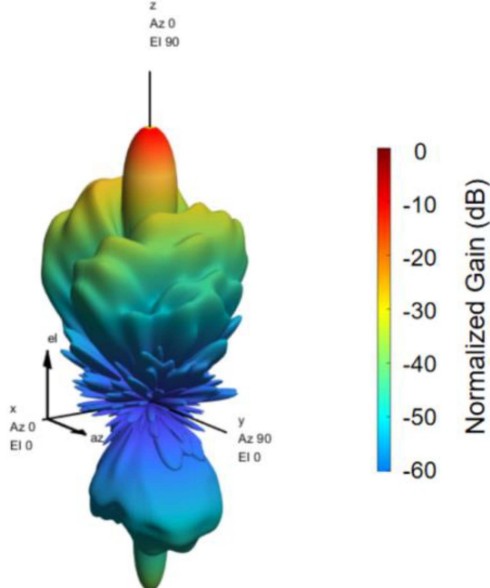

**Figure 12.** The 3D radiation pattern of the 512-element sparse array synthesized by MCDE algorithm.

### 3.3. Comparison with Existing Research

In this part, some studies based on sparse conformal arrays are compared with the proposed method, including PSO-SOCP, hybrid IWO/PSO, and the Compressed-Sensing Inspired Deterministic Algorithm. Commonly, these researches achieve good PSLL with a larger sparse rate by optimizing the excitation and phase of elements. Compared to the above-mentioned arrays, the proposed array can achieve radiation on a surface with a greater slope (Table 7). However, since the method is based on evolutionary algorithms, the performance is relatively lower than compressed-sensing optimization methods.

**Table 7.** The Array in Recent Literature and the Proposed Array.

| Ref. | Optimization Method | Sparse Ratio | PSLL (dB) | HPBW (°) |
|---|---|---|---|---|
| [29] | PSO-SOCP | 48.4% | −20.68 | 15 |
| [33] | Hybrid IWO/PSO | / | −20 | 10 |
| [47] | Sparse forcing synthesis | 50% | −20 | 40 |
| [15] | Compressed-sensing inspired deterministic algorithm | 77% | −23 | 40 |
| This work | MCDE | 56.8% | −21.56 | 15 |

## 4. Fabrication Analysis

Due to the higher operating frequency (15–17 GHz) of the array, the machining accuracy has a certain influence on the performance. Two aspects of fabrication analysis, array arrangement and element antenna structure are investigated in the following section.

### 4.1. Array Arrangements

For the machining accuracy of the array arrangement, we simulate the influence on the synthesized results by randomly changing the element position within a certain range and changing the element positions as a whole.

For example 1, we add random values within the range of [0–0.5 mm], [0–1 mm] to the three directions of x, y, and z on the basis of the element position obtained by MCDE optimization, and then synthesize the corresponding radiation patterns. The radiation patterns including different position variation ranges and the original MCDE optimization results are shown in Figure 13a. It can be seen that random movement in the range of [0–0.5 mm] has little impact on the radiation pattern, with the PSLL increasing by less than 0.1 dB. However, when the moving range is [0–1 mm], the pattern deteriorates to a certain extent with the PSLL increase of 1.5 dB. Therefore, we consider 0.5 mm as a threshold for machining error.

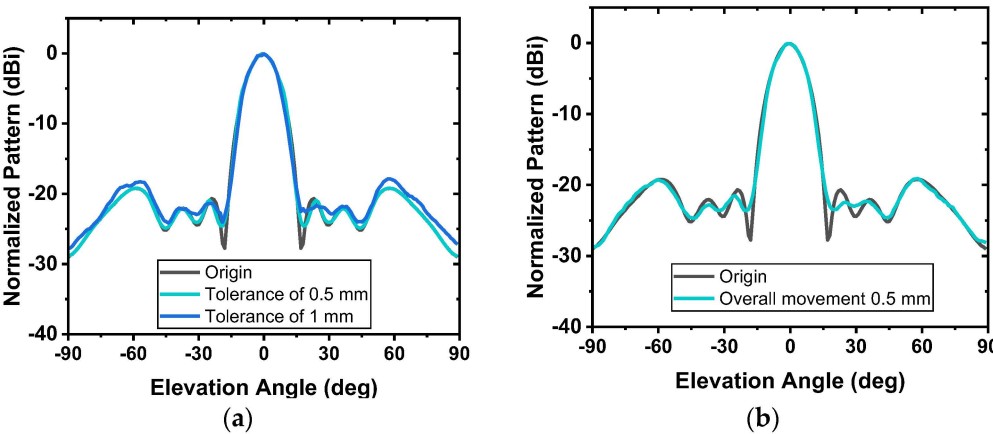

**Figure 13.** Normalized radiation patterns. (**a**) Sparse array synthesized by MCDE and array with different tolerances. (**b**) Normalized radiation patterns of the original sparse array and array with the overall movement of 0.5 mm.

Moreover, the array after a movement of 0.5 mm along the x-direction is synthesized and compared with the original sparse array in Figure 13b. It can be seen that the main beam of the array will not deviate and the PSLL increase is less than 0.2 dB.

### 4.2. Antenna Element

Another effect of array radiation is the fabrication of antenna elements. The metal of the patch antenna often has high machining accuracy and will not affect the radiation of the antenna. Thus, the processing difficulty of the element is mainly concentrated in the

position of the feeding structure. Therefore, we move the two feeding structures of the antenna along both the x and y directions. The simulation results show that the radiation of the antenna has little effect when the feeding structures move within 0.4 mm, as shown in Figure 14.

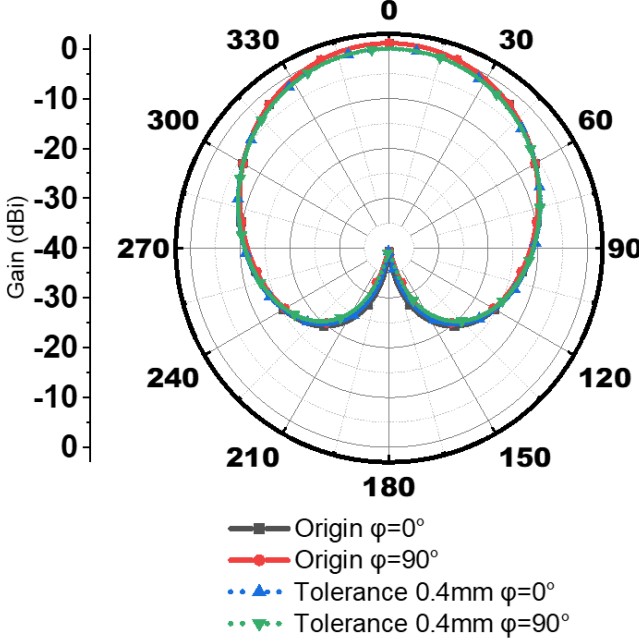

**Figure 14.** Normalized radiation patterns of original element and element with feeding structures' movement.

Through our research, a printed circuit board (PCB) is mostly used for the fabrication of such patch array antennas, and the mature process technology ensures that the machining accuracy of 0.4 mm is completely feasible at a low cost. Therefore, we believe that the arrays optimized by MCDE working in the 15–17 GHz range can be fabricated through basic processes.

## 5. Conclusions

A hybrid optimization method for the synthesis of the sparse conformal array with verification of a truncated cone antenna array is studied in this paper. By integrating a multi-agent system (MAS) with a differential evolution (DE) algorithm, a multi-agent composite differential evolution (MCDE) algorithm is designed to improve axial radiation and reduce peak sidelobe level (PSLL). Two antenna arrays in different sizes are proposed as examples and verify the effectiveness of the method. It is believed that MCDE will have a wide range of applications in the synthesis of conformal sparse arrays.

**Author Contributions:** Conceptualization, N.Z. and Z.X.; methodology, N.Z. and L.G.; software, N.Z., P.Z., and L.G.; validation, N.Z. and P.Z.; formal analysis, N.Z. and P.Z.; investigation, L.G.; resources, J.L.; data curation, N.Z. and J.L.; writing—original draft preparation, N.Z.; writing—review and editing, N.Z. and Z.X.; visualization, P.Z. and L.G.; supervision, J.L.; project administration, N.Z. and J.L.; funding acquisition, N.Z. and Z.X. All authors have read and agreed to the published version of the manuscript.

**Funding:** This research received no external funding.

**Data Availability Statement:** Not applicable.

**Conflicts of Interest:** The authors declare no conflict of interest.

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
