# Peer review of "Synthesis of Low Sidelobe Pattern with Enhanced Axial Radiation for Sparse Conformal Arrays Based on MCDE Algorithm"

_electronics, doi:10.3390/electronics11223679_

Round 1

Reviewer 1 Report

The solution proposed in this article is interesting. At a first glance the article could be published in this form. However, it contains several style errors, such as: 

1. The last pictures of page 11 are not counted;

2. It contains Figure 12 counted twice.

Otherwise, the study looks fine from my perspective, even though I recommend a more elaborate study. For instance, it is not clear how the implemented antenna looks like , maybe some readers would be interested to identify some details about connectivity, i.e. how the signal is fed to all these elements. 

Author Response

Thanks again for your valuable comments and the time you spend on this review, hereby our highest respect!

Reviewer 2 Report

The authors presented a hybrid optimization method for the synthesis of the sparse conformal array with verification of a truncated cone antenna array. The work is well presented with good analysis, however, the reviewer has following comments need to be addressed:

- The authors mainly considered suppressing the side lobes, but how the algorithm can be optimized for mutual coupling of the array elements and scan angles.

-What substrates (substrate 1, 2, and 3) are used for the proposed antenna?

- This reviewer feels that the conclusion should be rewritten more concisely and relevantly. Also, the authors mentioned "this letter" in the conclusion section; however, the article is submitted in a journal. The reviewer suggests the authors to be  mindful of these kind of mistakes while modifying a previous submission.

Author Response

Response to Reviewer 2 Comments

Point 1: The authors mainly considered suppressing the side lobes, but how the algorithm can be optimized for mutual coupling of the array elements and scan angles.

Response 1: Thanks for your comment. It is a good idea to optimize mutual coupling of the array elements and scan angles, while the algorithm in this research is used to suppress the side lobes and increase axial radiation. Based on the application of missile's data transmission, the synthesis aim here is to improve the axial radiation, so the scan angle is not considered as an optimization goal. Besides, we used the active element pattern to replace the traditional ideal omnidirectional pattern to increase the accuracy of the pattern synthesis by taking mutual coupling between elements into consideration. In future research we will explore methods to optimze mutual coupling between elements and scan angles.

Point 2: What substrates (substrate 1, 2, and 3) are used for the proposed antenna?

Response 2: Thanks for your comment, we apologize for not clarifying the substrstes. Rogers 5880 () is used as substrates in this antenna. We have clarified this information in the revised manuscript.

Point 3: This reviewer feels that the conclusion should be rewritten more concisely and relevantly. Also, the authors mentioned "this letter" in the conclusion section; however, the article is submitted in a journal. The reviewer suggests the authors to be  mindful of these kind of mistakes while modifying a previous submission.

Response 3: Thanks for your comment, we apologize for the mistakes in the manuscript. We have rewritten the manuscript and proofread the whole manuscript to avoid these kind of mistakes.

Revised manuscript:

  1. Conclusion

A hybrid optimization method for the synthesis of the sparse conformal array with verification of a truncated cone antenna array is studied in this paper. By integrating a multi-agent system (MAS) with a differential evolution (DE) algorithm, a multi-agent composite differential evolution (MCDE) algorithm is designed to improve axial radiation and reduce peak sidelobe level (PSLL). Two antenna arrays in different sizes are proposed as examples and verify the effectiveness of the method. It is believed that MCDE will have a wide range of applications in the synthesis of conformal sparse arrays.

Thanks again for your valuable comments and the time you spend on this review, hereby our highest respect!

Reviewer 3 Report

In the paper, the MCDE algorithm is used for synthesis of low-sidelobe pattern for sparse conformal arrays. The comments are listed below:

1. How to obtain the array factor of equation (1)? 

2. Since the antenna element is conformal to the cone, the simulated results for conformal antenna element is more suitable for importing into Matlab.

3. The accuracy of the optimized parameters should be discussed. For example, the optimized sparse conformal array is simulated in HFSS for validation.

Author Response

(The authors gave the same response as above.)

Round 2

Reviewer 2 Report

All the comments are addressed. I recommend to accept the article in its current form.

Reviewer 3 Report

The paper is accepted in its current version.